# Perilipin Isoforms and PGC-1α Are Regulated Differentially in Rat Heart during Pregnancy-Induced Physiological Cardiac Hypertrophy

**DOI:** 10.3390/medicina58101433

**Published:** 2022-10-11

**Authors:** Jesús A. Rosas-Rodríguez, Adolfo Virgen-Ortíz, Enrico A. Ruiz, Rudy M. Ortiz, José G. Soñanez-Organis

**Affiliations:** 1Departamento de Ciencias Químico-Biológicas y Agropecuarias, Universidad de Sonora, Lázaro Cárdenas del Río No. 100, Francisco Villa, Navojoa CP 85880, Mexico; 2Centro Universitario de Investigaciones Biomédicas, Universidad de Colima, Colima 28040, Mexico; 3Departamento de Zoología, Escuela Nacional de Ciencias Biológicas, Instituto Politécnico Nacional, Ciudad de México 07738, Mexico; 4Department of Molecular & Cell Biology, University of California Merced, Merced, CA 95343, USA

**Keywords:** perilipin, lipid metabolism, pregnancy, gene expression, protein abundance, PGC-1α

## Abstract

*Background and Objectives*: Perilipins 1–5 (PLIN) are lipid droplet-associated proteins that participate in regulating lipid storage and metabolism, and the PLIN5 isoform is known to form a nuclear complex with peroxisome proliferator-activated receptor-gamma coactivator 1-alpha (PGC-1α) to regulate lipid metabolism gene expression. However, the changes in PLIN isoforms’ expression in response to pregnancy-induced cardiac hypertrophy are not thoroughly studied. The aim of this study was to quantify the mRNA expression of PLIN isoforms and PGC-1α along with total triacylglycerol (TAG) and cholesterol levels during late pregnancy and the postpartum period in the rat left ventricle. *Materials and Methods*: Female Sprague-Dawley rats were divided into three groups: non-pregnant, late pregnancy, and postpartum. The mRNA and protein levels were evaluated using quantitative RT-PCR and Western blotting, respectively. TAG and total cholesterol content were evaluated using commercial colorimetric methods. *Results*: The expression of mRNAs for PLIN1, 2, and 5 increased during pregnancy and the postpartum period. PGC-1α mRNA and protein expression increased during pregnancy and the postpartum period. Moreover, TAG and total cholesterol increased during pregnancy and returned to basal levels after pregnancy. *Conclusions*: Our results demonstrate that pregnancy upregulates differentially the expression of PLIN isoforms along with PGC-1α, suggesting that together they might be involved in the regulation of the lipid metabolic shift induced by pregnancy.

## 1. Introduction

Perilipins (PLINs) are a family of structural proteins associated with the surface of intracellular lipid droplets (LDs) [1]. PLINs are involved in lipid metabolism and storage by regulating neutral fat (triacylglycerol—TAG—and cholesterol ester) contained in the core of LDs [2]. In mammals, five genes encode the five isoforms PLIN1, PLIN2 (ADRP, ADFP, or adipophilin), PLIN3 (Tip47, PP17, or M6PRBP), PLIN4 (S3–S12), and PLIN5 (MLDP, OXPAT, LSDP5, or Pat1), which differ in function, size, and affinity for LDs, tissue expression, transcriptional regulation, and unbound stability [3,4,5,6]. PLIN2 and PLIN3 are ubiquitously expressed across tissues [7,8]. PLIN1 and PLIN4 are abundantly expressed in adipose tissue (white and brown) [9,10], while PLIN5 is mainly expressed in cardiac and skeletal muscle [11,12].

PLIN isoforms contribute differently to various human diseases associated with impaired lipid metabolism, including cardiovascular diseases. PLIN1 deficiency in adipose tissue causes hypertension [13] and hypertrophic cardiomyopathy [14]. Other studies show that PLIN2 expression is increased in different cardiomyopathies [15], while PLIN3 expression increases with atherosclerosis [16]. PLIN5 can protect the heart from oxidative damage by sequestering lipids in LDs in the heart [17,18], forming barriers to prevent uncontrolled mobilization of TAG and excessive presence and oxidation of fatty acids (FA) [17]. This LD-mediated mechanism of lipid sequestration is proposed to protect the heart during diabetic cardiomyopathy [18].

Both pregnancy and exercise can induce physiological cardiac hypertrophy that is physiological, where the heart’s function is normal or enhanced [19]. On the other hand, hypertension, valve disease, genetic mutations, and other cardiovascular complications, induce maladaptive remodeling and metabolic cascades that generate pathological cardiac hypertrophy [20]. Normally, FA oxidation represents the main fuel for the adult heart. During pregnancy, this is enhanced by increasing the cardiac oxidative capacity, mitochondrial enzyme activity, and lipid metabolism [21]. In contrast, pathology-induced cardiac hypertrophy is characterized by a detrimental increased reliance on glucose and decreased utilization of fatty acids [22]. During pregnancy, the heart increases its circulating FA and TAG uptake to meet the increased metabolic demand [23]. When FA absorption overcomes its oxidation rate, excess FA is stored as TAG inside LDs, driving the ectopic storage of lipids [10]. During intensive exercise, intramuscular lipids increase, which changes the expression of PLINs and their association to mitochondria [24], a behavior that may be present in the heart during pregnancy due to increased FA uptake [25].

Moreover, it is known that PLIN5 can forms a transcriptional complex with peroxisome proliferator-activated receptor gamma coactivator 1-alpha (PGC-1α) to promote FA catabolism and mitochondrial gene biogenesis [26]. This furthers the impact of changes in PLIN5 expression and its relation to substrate metabolism regulation [27].

The participation of PLIN isoforms in lipid metabolism and storage during hypertrophic cardiomyopathies has been well described [13,14,15,16,17,18]. Furthermore, we have demonstrated that cardiac metabolites and key enzymes in lipid biosynthesis increase during pregnancy, demonstrating changes in lipid metabolism [28]. Nonetheless, there is a lack of attention given to the participation of PLIN isoforms and their association with LDs in response to pregnancy-induced cardiac hypertrophy. Therefore, we quantified the PLIN isoforms’ mRNA expression, as well as PGC-1α mRNA expression and nuclear protein abundance, along with TAG and cholesterol in the left ventricle during pregnancy and the postpartum period. Obtaining this data will further the recent efforts on understanding the physiological reprograming of the heart during pregnancy and the implications of changes in the expression of PLINs.

## 2. Material and Methods

### 2.1. Animal Handling

All procedures were reviewed and approved by the Bioethics Committee of the University of Colima, following the Guide for the Care and Use of Laboratory Animals (National Research Council, Washington, DC, USA).

Female Sprague-Dawley rats, three months old, were kept in groups of 2; their reproductive cycles were followed by vaginal smears [29], and animals were mated at appropriate times [30]. We have previously demonstrated the degree of physiological hypertrophy developed during pregnancy, and hence the pregnancy stages for correct group differences were divided as follows (Virgen-Ortiz et al., 2009). Rats were separated into non-pregnant as control (NP, diestrus 260 ± 8 g, *n* = 5), late pregnancy (LP, 18–21 days of gestation, 330 ± 10 g, *n* = 5), and postpartum (PP, 7 days after gestation, 270 ± 8 g, *n* = 5). Animals were provided with water and food (Lab Rodent Diet 5001) ad libitum. The individual cages were maintained on a 12:12 h light–dark cycle at a room temperature of 22 ± 1 °C and a humidity of 40–70%. For dissection, rats were anesthetized intraperitoneally (50 mg sodium pentobarbital/kg), and cervical dislocation was then performed. Each hearts was rapidly removed, and the left ventricle was dissected and stored at −80 °C.

### 2.2. Quantification of PLINs and PGC-1α Genes

Total RNA from the left ventricle was isolated using TRIzol reagent (Invitrogen, Waltham, MA, USA). Its integrity was evaluated by 260/280 nm ratio absorbance and agarose gel electrophoresis. Genomic DNA was degraded using DNase I (Roche, Indianapolis, IN, USA), and cDNA was synthetized using 2.5 μg total RNA with the GoScript^TM^ Reverse Transcriptase kit (Promega, Madison, WI, USA) and oligo-dT. Quantitative PCR was performed for all PLIN isoforms and PGC-1α, with primers designed based on their homology for the rat, *Rattus norvegicus* (Table 1). All PCR products were sequenced, and their identity was confirmed with their homology for *R. norvegicus*. Two PCR reactions for each cDNA were run using the Step-One Real Time PCR System (Applied Biosystems, Foster City, CA, USA) in a final volume of 15 μL with 125 ng of equivalent total RNA. After an initial denaturing step at 94 °C for 10 min, amplifications were performed for 40 cycles at 94 °C for 15 s and 63 °C for 1 min, and a final melting curve program increased 0.3 °C each 20 s from 60 to 95 °C. Negative controls were included for each gene. The mRNA concentration was obtained using standard curves from dilutions from 5 × 10^−4^ to 5 × 10^−8^ ng μL^−1^ of each PCR fragment and normalized to the expression of α-actin.

### 2.3. Western Blot Analysis and Stain-Free Technology Normalization

Nuclear protein was extracted from frozen left ventricles using an NE-PER extraction kit (Thermo Scientific, Waltham, MA, USA), and its concentration was evaluated using the Bradford protein assay (Bio-Rad, Hercules, CA, USA). Twenty µg of nuclear proteins was separated on TGX-Stain-Free gels (Bio-Rad) for 50 min at 120 V and transferred to a PVDF membrane using the Trans-Blot Turbo Transfer System (Bio-Rad). Membranes were blocked for 1 h with Blotting-Grade Blocker (Bio-Rad) and incubated 18 h at 4 °C with the anti-PGC-1α antibody (dilution 1:500) (Santa Cruz Biotechnology, Dallas, TX, USA) diluted in TBS-T 1% casein. Next, each membrane was washed with TBS-T and incubated for 1 h with its respective secondary antibody (dilution 1:20,000) containing 1% casein. A chemiluminescent signal was captured following the instructions for the Clarity™ Western ECL Blotting Substrates (Bio-Rad). Finally, the data were analyzed according to Western Blot Normalization with Image Lab Software guide, and the linear range of the signal was evaluated following the methods previously reported [31].

### 2.4. Triacylglycerol and Cholesterol Ester Measurement

Total TAG and cholesterol concentration were measured in left ventricles using the Triglyceride Assay Kit—Quantification (Abcam, Cambridge, UK) and Cholesterol/Cholesteryl Ester Detection Kit (Abcam), respectively, following the manufacturer instructions. All measurements were performed in triplicate, measuring the absorbance of each sample at 570 nm in the Varioskan LUX multimode microplate reader (Thermo Scientific). Results were obtained as mg/dL and were further normalized with mg of tissue.

### 2.5. Statistical Analysis

Data were expressed as means ± SEM and tested for normality with the D’Agostino–Pearson test. The data were compared by one-way ANOVA analysis of variance and Tukey’s post-hoc test. Values are reported as means ± SD, and statistically significant differences were considered at *p* < 0.05. Statistical analysis was conducted using the Minitab Statistical Software release version 12 (Minitab, LLC, State College, PA, USA).

## 3. Results

### 3.1. PLIN Isoforms Are Differentially Regulated during and after Pregnancy

In NP, PLIN5 mRNA was the most expressed transcript, followed by PLIN1, PLIN3, and PLIN2, while PLIN4 was not detected under any condition (Figure 1). During LP and PP, PLIN1 mRNA expression increased 9.5- and 7.4-fold, respectively, compared to NP (Figure 2A). Expression of PLIN2 increased 28- and 18-fold in LP and PP, respectively, compared to NP (Figure 2B). During PP, PLIN2 mRNA was 1.5-fold lower than in LP (Figure 2B). Significant changes in PLIN3 expression were not observed during any conditions (Figure 2C). Expression of PLIN5 increased 8- and 2.4-fold during LP and PP, respectively, compared to NP (Figure 2D). During PP, PLIN5 mRNA was 3-fold lower than in LP (Figure 2D).

### 3.2. mRNA and Protein Expression of PGC-1α Increased during and after Pregnancy

PGC-1α mRNA levels increased 18- and 11-fold during LP and PP respectively, compared to NP (Figure 3A). During PP, PGC-1α expression decreased 1.6-fold compared to LP. The PGC-1α protein abundance was evaluated using the Stain-Free technology as a normalization tool in a Western blot analysis (see Appendix A) [31]. The result indicates that PGC-1α protein abundance increased 1.5- and 1.2- fold in LP and PP, respectively, compared to NP (Figure 3B).

### 3.3. Cardiac Total TAG and Cholesterol Concentration Increased during Pregnancy

Cardiac total TAG and cholesterol increased 1.5-fold in LP compared to NP and returned to basal levels during PP (Figure 4).

## 4. Discussion

This study demonstrates that pregnancy and postpartum status increase the mRNA expression of the PLIN1, 2, and 5 isoforms. Furthermore, we showed that the mRNA expression and nuclear content of PGC-1α increases during pregnancy and the postpartum period, along with cardiac total TAG and cholesterol content.

PLIN5 is the most expressed PLIN isoform during non-pregnancy conditions, followed by PLIN1, 2, and 3. Interestingly, PLIN4 expression was not detected in samples under any conditions. Pregnancy increased the expression of PLIN1, 2, and 5, suggesting that these may be required for LD biogenesis [7] and the regulation of LD content metabolism [10]. The increase in PLIN1 and 2 during pregnancy and the postpartum period may facilitate LD formation and expansion. This would allow for increased storage of excess TAG and FA associated with the increased metabolic demand driven by pregnancy [23]. In this study, PLIN5 expression and lipids (i.e., total TAG and cholesterol) increase in the heart during pregnancy, suggesting that PLIN5 may participate in regulating lipid metabolism, preventing lipids’ release from LD and limiting FA toxicity [32].

The function of the ubiquitously expressed PLIN3 is not well understood [10]. PLIN3 may contribute to LD biogenesis in neutrophils [33], regulate lipid metabolism and thermogenic gene expression in beige adipocytes [34], and promote lipid oxidation in *vastus lateralis* muscle [35]. It is also suggested that it may compensate for the lack or loss of PLIN2 or PLIN5 in specific tissue [10]. However, PLIN5 was the most dynamically expressed isoform in our study [36]. This suggests that PLIN5 may serve as the primary isoform regulating lipid metabolism in the heart during the stages of pregnancy and postpartum status studied here. While PLIN3 increases with newly formed LDs [9], in this study, the period for nascent LD formation may be during early pregnancy [23,37]; instead, the data we captured reflect established pregnancy, supporting the lack of change in PLIN3 expression.

PLIN5 serves as a metabolic modulator of cardiac LDs [38] and is highly expressed in hypertrophic cardiomyopathy [39]. PLIN5 expression is increased in the heart and skeletal muscle during conditions that promote increased FA metabolism (i.e., fasting and exercise) [38]. In PLIN5-deficient mice, cardiac LDs were undetectable [40], potentially exposing the heart to increased oxidative damage [41]. This is especially important because we have shown that FA availability increases during pregnancy in these rats [28]. In this study, PLIN5 expression and lipid contents (total TAG and cholesterol) increase during pregnancy, suggesting that PLIN5 may participate in lipid storage. This prevents TAG release and decreases FA oxidation, reducing the potential for FA toxicity [17,18].

The expression of PLIN2 and PLIN5 decreased in the postpartum period. This may be supported by the decrease in cardiac total TAG and cholesterol we detected, as the PLIN2 and 5 isoforms participate in LD formation and cardio-protection, respectively [7,42]. These results highlight a possible role for PLIN expression in cardiac substrate metabolism induced by pregnancy. PLINs increase during higher metabolic demand and decrease during the postpartum period, following the return of the heart to normal conditions [23,43]. 

PGC-1α is the master regulator of mitochondrial biogenesis and respiration. In pregnant rats, PGC-1α is increased [44], while PGC-1α-null mice experienced metabolic deficiency [25]. In our study, PGC-1α expression and nuclear protein abundance increased during pregnancy and decreased in the postpartum period. This suggests that PGC-1α participates in allowing the heart to meet the increased demand for ATP during pregnancy when cardiac function is higher [21,25,44]. Furthermore, PGC-1α upregulation starts to reverse during the postpartum period, when the metabolic demand decreases [25]. Additionally, PLIN5 can form transcriptional nuclear complexes with PGC-1α and sirtuin 1 (SIRT1) [26], which can promote PGC-1α functioning [27]. Also, the transcriptional activity of peroxisome proliferator-activated receptor γ (PPARγ) increases in the left ventricle during pregnancy and the postpartum period [28]. Our data show that both PLIN5 and PGC-1α expression and nuclear abundance are elevated during conditions of increased metabolic demand (i.e., pregnancy). However, more studies are needed to fully understand the PLIN family and its role in regulating lipid metabolism.

## 5. Conclusions

The present study demonstrates that PLIN isoforms are differentially expressed during and after pregnancy and parallel changes in PGC-1α and total cardiac TAG content, suggesting that they may participate in the lipid metabolic shift induced by pregnancy. This study demonstrated that all but PLIN4 isoforms were detectable in the heart. We identified PLIN5 and PLIN2 to be the most dynamically expressed PLIN isoforms. This suggests that these isoforms may contribute the most to the regulation of cardiac lipid metabolism during pregnancy. Because pregnancy induces reversible cardiac hypertrophy, it presents the opportunity to study the heart during a physiological adaptation. Studying this state provides insight into reversing the adverse metabolic effects of pathology-induced cardiac hypertrophy.

## Figures and Tables

**Figure 1 medicina-58-01433-f001:**
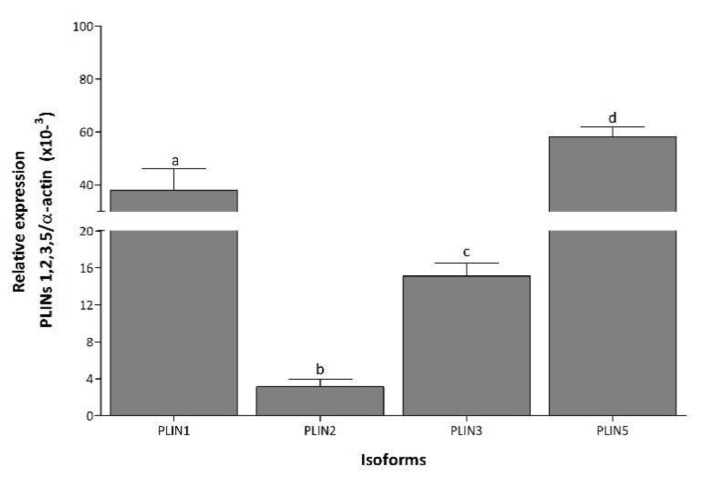
Differential expression of PLIN isoforms in left ventricle. Mean (±S.E.M.) mRNA expression levels of PLIN1, 2, 3, and 5 in left ventricle of non-pregnant rats. Values with the same letter are not significantly different (*p* < 0.05).

**Figure 2 medicina-58-01433-f002:**
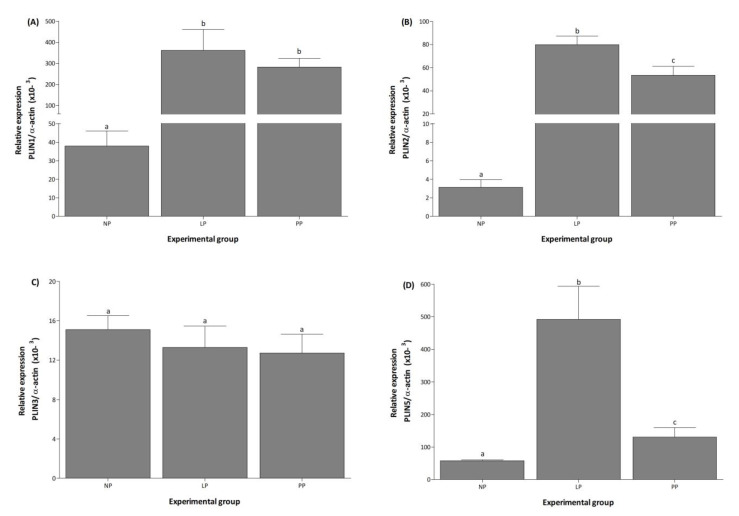
Pregnancy induces the expression of PLIN1, PLIN2, and PLIN5 in the left ventricle. Mean (±S.E.M.) mRNA expression levels of PLIN1 (**A**), 2 (**B**), 3 (**C**), and 5 (**D**) in the left ventricle of non-pregnant (NP, *n* = 5), late pregnancy (LP, *n* = 5), and postpartum (PP, *n* = 5) rats. Different letters indicate significant differences (*p* < 0.05) between the NP group and the LP and PP groups.

**Figure 3 medicina-58-01433-f003:**
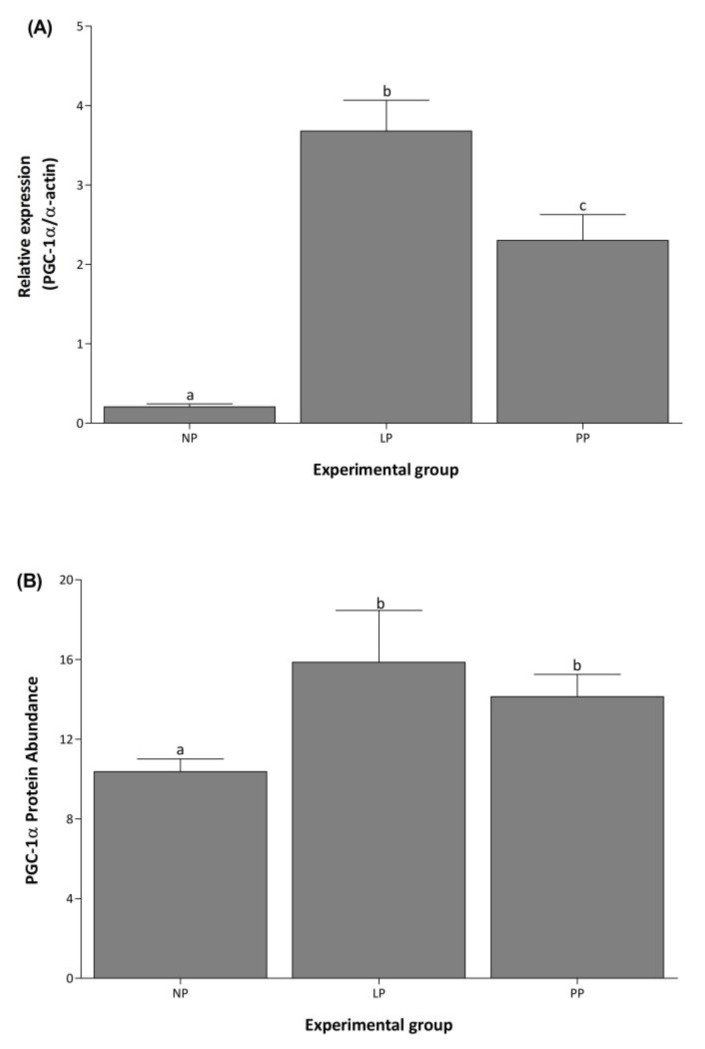
PGC-1α increases in the left ventricle during pregnancy. Mean (±S.E.M.) mRNA expression (**A**) and protein abundance (**B**) of PGC-1α in the left ventricle of non-pregnant (NP, *n* = 5), late pregnancy (LP, *n* = 5), and postpartum (PP, *n* = 5) rats. Different letters indicate significant differences (*p* < 0.05) between the NP group and the LP and PP groups.

**Figure 4 medicina-58-01433-f004:**
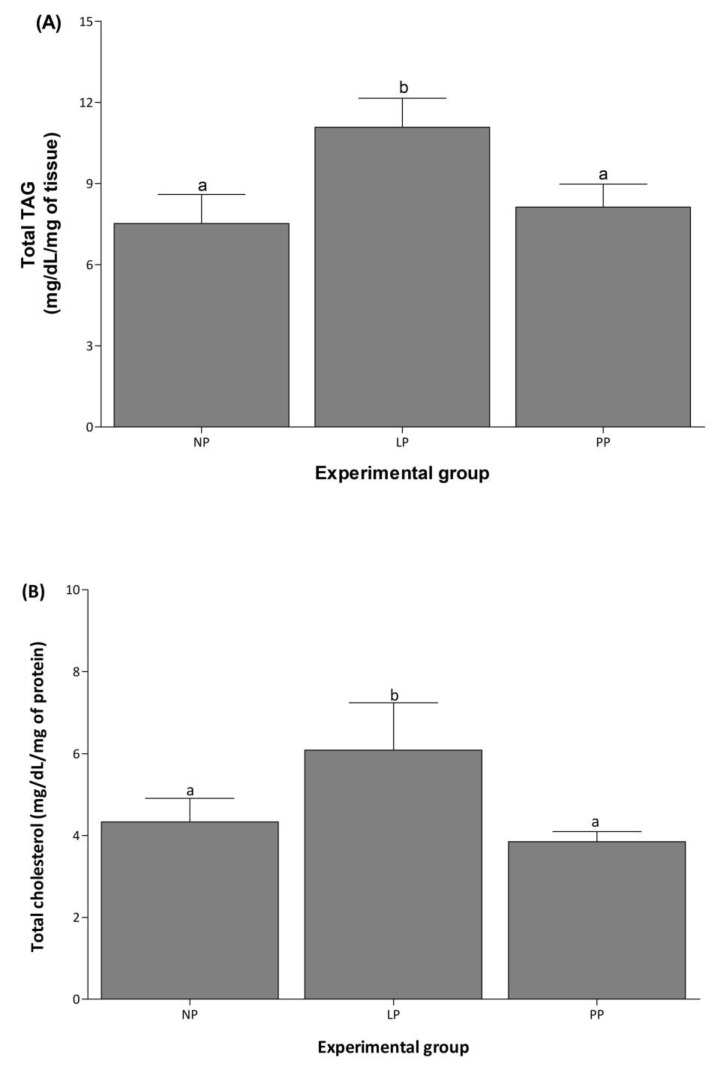
Cardiac total TAG and cholesterol increase during pregnancy. Mean (±S.E.M.) of total TAG (**A**) and cholesterol (**B**) contents in left ventricle of non-pregnant (NP, *n* = 5), late pregnancy (LP, *n* = 5), and postpartum (PP, *n* = 5) rats. Different letters indicate significant differences (*p* < 0.05) between the NP group and the LP and PP groups.

**Table 1 medicina-58-01433-t001:** Primers used for the quantitative PCR.

Primer Name	Nucleotide Sequences (5′-3′)	GenBank AccessionNumber	Product Size (Base Pairs)
*PLIN1F1* *PLIN1R2*	GAGTCACAACCCCACGATGT CGAGAGAGGAAAGAGTCGAC	NM_001308145.1	119
*PLIN2F1* *PLIN2R2*	CAGTACTTGCCGCTCACTCA TCAGATGGACAGTGGAGTGG	NM_001007144.1	214
*PLIN3F2* *PLIN3R2*	CTCAGTGTCTGGTGCAAAGG CTAGTTCTGTGTCTGTCAGG	XM_236783.6	233
*Plin4F1* *PLIN4R1*	CCGAGTACTCTCTGCCAACC AATCTGCCACCTTGCATCCC	XM_006244373.3	218
*PLIN5F2* *PLIN5R2*	CCAAGCCATGGACACTGTGC GCCGATAGGGATCCCAGACG	NM_001134637.1	210
*PGC-1αF1* *PGC-1αR1*	TTCTTCCACAGATTCAAGCC CATTACTGAAGTTGCCATCC	AB025784.1	210
*aActinF1* *aActinR1*	ATGTGTGACGACGAGGAGACC CTACATAGGAGTCTTTCTGCC	X80130.1	169

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
