# Peer review of "Perilipin Isoforms and PGC-1α Are Regulated Differentially in Rat Heart during Pregnancy-Induced Physiological Cardiac Hypertrophy"

_medicina, 2022, doi:10.3390/medicina58101433_

Round 1
Reviewer 1 Report
In the submission entitled “Perilipin isoforms and PGC-1α are regulated differentially in rat heart during pregnancy induced physiological cardiac hypertrophy” Rodríguez et al studied the interplay of PLIN isoforms i.e., and PGC-1α and their impact on lipid metabolism in late pregnancy and postpartum in animal model (rat). They assessed gene expression in rat left ventricle to focus on cardiac hypertrophy.
Following are the observations in the study:
Major:
A: This comparative study must include the protein expression data. Although authors repeatedly mentioned in the manuscript about protein expression and western blot e.g.,
1: in the Abstract lines 45-47
PLIN isoforms were 45 differentially regulated during and after pregnancy in the left ventricle, while PGC-1α 46 mRNA expression and protein were upregulated.
2: Methods, section Line #136-149
2.3. Western blot analysis and stain-free technology normalization
Nuclear protein was extracted from frozen left ventricle using NE-PER extraction kit
………Finally, the images were analyzed according to Western blot normalization with Image Lab Software guide, and the linear range of the signal was evaluated following the methods previously reported
Western blot images are missing as the authors only mentioned the densitometry graphic data.
Secondly PLIN isoforms protein expression data is totally missing.
B: Author has mentioned in section about cardiac TAG but in the Figure they have mentioned total TAG not cardiac?
3.3. Cardiac TAG and cholesterol concentration increased during pregnancy
Cardiac TAG and total cholesterol increased 1.5- fold in LP compared to NP and 191 returned to basal levels during PP (Fig. 4).
Minor
Line 81 and 82
During pregnancy, the heart increases its circulating FA and TAG uptake during pregnancy
Line 108
Female Sprague-Dawley rats (3 moth-old) were kept in groups of 2
Line 113
And food (Lab Rodent Diet 5001) ad libitum and maintained (sentence structure and spelling)
Line 197
The literals denote significant 197 changes (p <0.05) between each experimental group.
Author Response
We thank the reviewer for their careful review and insightful comments. We have considered their suggestions and incorporated the corrections accordingly. We would like to thank them for maintaining a thorough review standard that supports the quality of the research featured in Medicina.
Reviewer 1
Major:
A: This comparative study must include the protein expression data. Although authors repeatedly mentioned in the manuscript about protein expression and western blot e.g.,
1: in the Abstract lines 45-47
PLIN isoforms were 45 differentially regulated during and after pregnancy in the left ventricle, while PGC-1α 46 mRNA expression and protein were upregulated.
We appreciate the comment, the conclusion was rewritten.
2: Methods, section Line #136-149
2.3. Western blot analysis and stain-free technology normalization
Nuclear protein was extracted from frozen left ventricle using NE-PER extraction kit
………Finally, the images were analyzed according to Western blot normalization with Image Lab Software guide, and the linear range of the signal was evaluated following the methods previously reported
Western blot images are missing as the authors only mentioned the densitometry graphic data.
We appreciate the comment, we add as supplemental figure the PGC-1a western blot normalization analysis including the blot after cutting membrane; the electrophoresis and the stain-free analysis in Stain-Free SDS; the linear and correlation of the PGC-1a antibody fluorescent signal with sample concentration.
Secondly PLIN isoforms protein expression data is totally missing.
We understand the concern brought up by the reviewer, the various mentions of “expression” can indeed be usually associated with protein, not exclusively with genes. We have carefully revised our manuscript to properly reflect the data presented either for mRNA or protein changes.
Based on molecular biology literature, “expression” refers exclusively to “gene expression”, while for proteins the terms are broader, yet here we chose “abundance”, as we do a qualitative assessment of densitometry that is calculated to be equate protein abundance. In our manuscript, we use the distinction, when present data concerning mRNA levels, we refer to them as “expression” (e.g., line 170: “Significant changes in PLIN3 expression were not observed ...”), whereas for proteins we refer to it as “abundance” (e.g., line 188: “Accordingly, PGC-1α protein abundance increased …”).
B: Author has mentioned in section about cardiac TAG but in the Figure they have mentioned total TAG not cardiac?
3.3. Cardiac TAG and cholesterol concentration increased during pregnancy
Cardiac TAG and total cholesterol increased 1.5- fold in LP compared to NP and 191 returned to basal levels during PP (Fig. 4).
The denomination of “total” can be indeed confusing, as to where the measure was obtained from. We have changed our graph “Y" axis title and the associated mentions in the text to “Total cardiac triacylglycerol (TAG)”. Given the many species of TAG measured by the commercial kits, we chose to present our data as “total”, as it furthers the understanding of the presence of different species.
Minor
Line 81 and 82
During pregnancy, the heart increases its circulating FA and TAG uptake during pregnancy
Line 108
Female Sprague-Dawley rats (3 moth-old) were kept in groups of 2Line 113
And food (Lab Rodent Diet 5001) ad libitum and maintained (sentence structure and spelling)
Line 197
The literals denote significant 197 changes (p <0.05) between each experimental group.
We have corrected the text accordingly and thoroughly reviewed our manuscript before resubmission to ensure that errors of this type are not present.

Reviewer 2 Report
In the manuscript by Rosas-Rodriguez et al., the authors present a well-written study. They first characterize PLIN expression in the left ventricle of rats during and after pregnancy, followed by the investigation of PGC1-a expression and TAG and Total cholesterol levels during and after pregnancy. The authors claim that results suggest that PLIN isoforms and PGC1-a are involved in the regulation of the lipid metabolism shift induced by pregnancy and that this finding provides new insights into reversing the adverse metabolic effects of pathology-induced cardiac hypertrophy. Although interesting and well written, the novelty of the results is limited. Several concerns are particularly described below.
1. In line 260 the author states that the present study demonstrated that pregnancy-induced cardiac hypertrophy is associated with a dynamic change in PLINs and parallel changes in PGC1-a and cardiac TAG content. To strengthen these results the author should at least show that pregnant rats used in this study did develop pregnancy-induced cardiac hypertrophy. No results on hypertrophic hearts are provided, which makes this conclusion relatively weak. Cardiac function and morphometric changes should be assessed with echocardiography.
2. The authors mention a lipid metabolic shift induced by pregnancy (line 50). Yet only one parameter (TAG and cholesterol levels) is investigated that supports this claim. What happens to fatty acid utilization in these animals? More readouts regarding metabolic shift (for instance FAOX of glucose uptake) should be provided.
3. Although the authors show increased PGC1-a and PLIN (particularly PLIN5) mRNA expression levels in left ventricles of pregnant rats, only correlations with pregnancy are shown. The authors do not show any causal relationships between these PGC1-a, TAG, and cholesterol levels and PLIN expression levels. Therefore, authors should be very careful with their too strong conclusions. Expression levels of PGC1-a and TAG and cholesterol should be investigated in a pregnancy-induced cardiac hypertrophy model after modulation of PLIN5 expression in the heart (knock-out/PLIN inhibitors).
4. Next to PGC1-a mRNA expression, PGC1-a protein abundance was investigated. Although the authors did find a reduction in PGC1-a mRNA levels postpartum, no change in protein abundance was found. To strengthen results PLIN5 expression levels (mRNA and active protein (PLIN 5 S155 phosphorylation) should be measured). Moreover, it would be interesting to see what happened to upstream and downstream targets of PGC1-a. For instance, p-Akt, p-AMPK, p-PKA, p-HDAC, MEF-2, PPARs could be measured. Conclusions are now only based on increased PGC1-a mRNA levels.
5. In line 260 the author states that the present study demonstrated that pregnancy induced cardiac hypertrophy is associated with a dynamic change in PLINs and parallel changes in PGC-1a and cardiac TAG content. To strengthen these results the author should at least show that pregnant rats used in this study did develop pregnancy induced cardiac hypertrophy. No results on hypertrophic hearts are provided, which makes this conclusion relatively weak. Cardiac function and Morphometric changes should be assessed with echocardiography.
6. Why did the author choose or 7 days postpartum? Could it be that expression levels of PLIN1 and PLIN2 would return to NP if the postpartum period would have been prolonged?
7. For the WB only quantified data are shown. Please add raw WBs to the bars.
Line specific remarks:
- Line 93: Please add reference to this statement.
- Line 170: author mentions that during PP PLIN2 expression was 1.5-fold lower than NP. Based on the data it should be 18-fold higher compared to NP. I assume the author meant 1.5-fold lower than LP?
- Line 207: Be careful with choice of words. “The upregulation is reversible postpartum does not account for all the PLIN isoforms”. this sentence gives the impression that all isoforms are reversible postpartum.
- Line 216-219: “In this study…….limiting FA toxicity”. For your discussion and conclusion section it would be of value to provide a schematic figure of this hypothesis.
- Line 240: rephrase: “During postpartum, the expression of PLIN2 and PLIN5 decreased”.
- Line 241: Very strong statement. No causal relation between PLINs and cardiac TAG and cholesterol has been shown in this study.
- Line 243: Please attenuate to “These results highlight a possible role for PLIN expression in cardiac substrate metabolism induced by pregnancy”.
- Line 245: Be careful with choice of words. A decrease postpartum was only shown for PLIN5.
- Line 250: The term “crucial” is a too strong formulation based on just 1 change in mRNA level.
- Line 265: “may contribute the most to the regulation of cardiac lipid metabolism and cardio-protection during pregnancy”. Please rephrase this sentence, too strong conclusion based on results.
- Figure description mentions “the literals denote significant changes……between each experimental group”. The literals are confusing, please specify which comparison is made under a, which under b and which under c.
Author Response
Reviewer 2
- In line 260 the author states that the present study demonstrated that pregnancy-induced cardiac hypertrophy is associated with a dynamic change in PLINs and parallel changes in PGC1-a and cardiac TAG content. To strengthen these results the author should at least show that pregnant rats used in this study did develop pregnancy-induced cardiac hypertrophy. No results on hypertrophic hearts are provided, which makes this conclusion relatively weak. Cardiac function and morphometric changes should be assessed with echocardiography.
We have previously presented the data corresponding to the pregnancy progression and heart remodeling. With the idea to not repeat what has been presented before, we did not include the corresponding data. We believe the new and relevant data presented in our manuscript may stand as is, though accordingly, we have included the relevant references and brief explanations in the “Methods” section to properly support and substantiate the results and hypotheses presented in our manuscript.
- The authors mention a lipid metabolic shift induced by pregnancy (line 50). Yet only one parameter (TAG and cholesterol levels) is investigated that supports this claim. What happens to fatty acid utilization in these animals? More readouts regarding metabolic shift (for instance FAOX of glucose uptake) should be provided.
We appreciate the assessment presented by this reviewer for their inquiry. We have made reference to the work where the metabolic parameters have been included. We have revised our work to properly reference that work when discussing our new results.
- Although the authors show increased PGC1-a and PLIN (particularly PLIN5) mRNA expression levels in left ventricles of pregnant rats, only correlations with pregnancy are shown. The authors do not show any causal relationships between these PGC1-a, TAG, and cholesterol levels and PLIN expression levels. Therefore, authors should be very careful with their too strong conclusions. Expression levels of PGC1-a and TAG and cholesterol should be investigated in a pregnancy-induced cardiac hypertrophy model after modulation of PLIN5 expression in the heart (knock-out/PLIN inhibitors).
We understand that knockout or inhibitory studies can provide valuable insight to this study. We have revised our text, adjusting for the statements presented to be more closely supported by our results. In the present work, we suggest that PGC and perilipins may participate in the metabolic regulation of the reprogramming that presents during physiological cardiac hypertrophy. We make primarily use of the cited literature to substantiate those claims, as we understand our results alone may not fully address the metabolic reprogramming.
- Next to PGC1-a mRNA expression, PGC1-a protein abundance was investigated. Although the authors did find a reduction in PGC1-a mRNA levels postpartum, no change in protein abundance was found. To strengthen results PLIN5 expression levels (mRNA and active protein (PLIN 5 S155 phosphorylation) should be measured). Moreover, it would be interesting to see what happened to upstream and downstream targets of PGC1-a. For instance, p-Akt, p-AMPK, p-PKA, p-HDAC, MEF-2, PPARs could be measured. Conclusions are now only based on increased PGC1-a mRNA levels.
We appreciate this point. While the activation of perilipins has been greatly advanced in recent years, the extent and coverage of such studies may be further than the one intended for our work. Regarding the activation of Plin5 in the heart, the current literature captures perilipin 5 S155 phosphorylation in liver. We understand that the measure may be insightful for our study yet given the different timelines and current challenges derived from the pandemic, we find ourselves in a position where the suggested measurements are not feasible. We again support the suggestion of the reviewer, though we also feel a study that investigates deeper metabolic pathways would benefit from such measurements. Our study presented the genetic expression profile of the perilipins as foundational research for future studies investigating this metabolic reprogramming in the context of lipid droplets.
- In line 260 the author states that the present study demonstrated that pregnancy induced cardiac hypertrophy is associated with a dynamic change in PLINs and parallel changes in PGC-1a and cardiac TAG content. To strengthen these results the author should at least show that pregnant rats used in this study did develop pregnancy induced cardiac hypertrophy. No results on hypertrophic hearts are provided, which makes this conclusion relatively weak. Cardiac function and Morphometric changes should be assessed with echocardiography.
We understand the concern regarding the state of the cardiac remodeling in the animals used for this study. We kindly remark that, as previously mentioned in point #1 of this reviewer’s response, we have previously presented the data corresponding to the pregnancy progression and heart remodeling. Nonetheless, we included the relevant references and brief explanations in the “Methods” section to properly support and substantiate the results and hypotheses presented in our manuscript.
- Why did the author choose or 7 days postpartum? Could it be that expression levels of PLIN1 and PLIN2 would return to NP if the postpartum period would have been prolonged?
- For the WB only quantified data are shown. Please add raw WBs to the bars.
Along with our submission, the reviewers can find the supplementary file containing the Western blot raw images used for the protein quantification. As well, we have included the representative image corresponding to the values from the Western blot densitometry calculated by the ImageLab software.
Line specific remarks:
- Line 93: Please add reference to this statement.
We have added the corresponding citation.
- Line 170: author mentions that during PP PLIN2 expression was 1.5-fold lower than NP. Based on the data it should be 18-fold higher compared to NP. I assume the author meant 1.5-fold lower than LP?
We have corrected the text to correctly define the comparisons.
- Line 207: Be careful with choice of words. “The upregulation is reversible postpartum does not account for all the PLIN isoforms”. this sentence gives the impression that all isoforms are reversible postpartum.
We have modified our text to reflect the results presented.
- Line 216-219: “In this study…….limiting FA toxicity”. For your discussion and conclusion section it would be of value to provide a schematic figure of this hypothesis.
- Line 240: rephrase: “During postpartum, the expression of PLIN2 and PLIN5 decreased”.
We have fixed the wording of the sentence as suggested.
- Line 241: Very strong statement. No causal relation between PLINs and cardiac TAG and cholesterol has been shown in this study.
We understand the limitation of the work presented and have changed the text accordingly.
- Line 243: Please attenuate to “These results highlight a possible role for PLIN expression in cardiac substrate metabolism induced by pregnancy”.- Line 245: Be careful with choice of words. A decrease postpartum was only shown for PLIN5.
- Line 250: The term “crucial” is a too strong formulation based on just 1 change in mRNA level.
- Line 265: “may contribute the most to the regulation of cardiac lipid metabolism and cardio-protection during pregnancy”. Please rephrase this sentence, too strong conclusion based on results.
For lines 243, 245 and 250, we have modified our text accordingly.
- Figure description mentions “the literals denote significant changes……between each experimental group”. The literals are confusing, please specify which comparison is made under a, which under b and which under c.
We agree, and as suggested, this has been corrected in the text.

Round 2
Reviewer 1 Report
Dear José G. Soñanez-Organis , your submitted revised manuscript entitled
"Perilipin isoforms and PGC-1α are regulated differentially in rat heart during pregnancy-induced physiological cardiac hypertrophy"
covered all major objections.
Reviewer 2 Report
The authors amended their manuscript adequately to the comments of the reviewers. This version is clearly improved. The authors have revised their manuscript text to statements that are more closely supported by their results. Reference to their previous work to show the effect of pregnancy on physiological cardiac hypertrophy has been included. Raw WB data are provided in the supplementary. Personally I think it is a pity that metabolic pathways haven't been studies to a deeper extent. It could have been a fruitful addition to this paper. However, this will leave room for future research.